# Fermentative Production of Erythritol from Cane Molasses Using *Candida magnoliae*: Media Optimization, Purification, and Characterization

Shruthy Seshadrinathan * and Snehasis Chakraborty *

Department of Food Engineering and Technology, Institute of Chemical Technology, Mumbai 400019, India
* Correspondence: fbt18sg.seshadrinathan@ictmumbai.edu.in (S.S.); sc.chakraborty@ictmumbai.edu.in (S.C.)

**Abstract:** Erythritol is a natural, zero-calorie sweetener that can be used as a sugar substitute and humectant for different products such as confectionaries, tablets, etc. Methods such as extraction and chemical synthesis for erythritol synthesis are not feasible or sustainable due to lower yield and higher operating costs. In the present study, erythritol is produced through the submerged fermentation of cane molasses, a by-product of the cane sugar industry, in the presence of the osmophilic yeast *Candida magnoliae.* Erythrose reductase enzyme assay was used for quantifying erythritol yield. Plackett–Burman's design screened the three most influential factors viz. molasses, yeast extract, and $KH_2PO_4$ out of 12 contributing factors. Further, the concentration of molasses (200–300 g/L), yeast extract (9–12 g/L), and $KH_2PO_4$ (2–5 g/L) were optimized using response surface methodology coupled with numerical optimization. The optimized erythritol yield (99.54 $g \cdot L^{-1}$) was obtained when the media consisted of 273.96 $g \cdot L^{-1}$ molasses, 10.25 $g \cdot L^{-1}$ yeast extract, and 3.28 $g \cdot L^{-1}$ $KH_2PO_4$ in the medium. After purification, the liquid chromatography–mass spectrometry (LC-MS) analysis of erythritol crystals from this optimized fermentation condition showed 94% purity. Glycerol was produced as the side product (5.4%) followed by a trace amount of sucrose and mannitol. The molecular masses of the erythritol were determined through mass spectrometry by comparing [M + Na] + ions. Analysis in electrospray (ES) positive mode gave ($m/z$) of 145.12 [M + 23]. This study has reported a higher erythritol yield from molasses and used osmotolerant yeast *Candida magnoliae* to assimilate the sucrose from molasses.

**Keywords:** erythrose reductase; Plackett–Burman design; response surface methodology; numerical optimization; sweetener; osmophilic yeast

## 1. Introduction

Erythritol is a four-carbon first-generation polyol with a glycosidic nature formed by the hydrolysis of aldehyde or a ketone group present in carbohydrates [1,2] This symmetric molecule occurs in meso form, having a molecular weight of 122.12 g/mol—the lowest among all sugar alcohols. It acts as a bulking sweetener. It is 60–80% sweet as sucrose. However, it does not contribute to calories, unlike sucrose [1]. According to the Food Safety and Standards Authority of India (FSSAI) regulation, erythritol is the only polyol providing zero calories, whereas other polyols contribute to 2 kcal/g. It shows antioxidative, non-carcinogenic, nonglycemic, high digestive tolerance, and non-acidogenic properties [3]. In 2019, the worldwide erythritol market was more than 195 million USD, and it is expected to grow at the rate of 6.5% CAGR from 2020 to 2026 [4]. Fermentative production of erythritol can fulfill this demand as this is selective and economical [5,6]. A set of various carbon sources has been used to produce erythritol through microbial fermentation, such as glycerol [7–9] monosaccharides [10], xylose [11], and molasses [12]. The various types of biological production of erythritol have been summarized by Rzechonek et al. [13], where sources such as molasses, glucose, and glycerol have been used. Molasses is a dark and dense final effluent obtained after repeated crystallization of sugarcane juice for sugar

production. It is mainly used as a carbon source for producing baker's yeast, citric acid, feed yeasts, acetone/butanol, organic acids, amino acids, antibiotics, and enzymes [14]. Molasses mainly contains non-reducing sugars (sucrose 30–40%), reducing sugars (10–25%), and trace minerals (7–15%) [15]. Hence, molasses could be used as a carbon substrate to produce sweeteners, which not only utilizes and reduces waste but also helps in reducing the carbon footprint [16]. For instance, Mirończuk et al. [12] used molasses as a raw material for erythritol production using *Yarrowia lipolytica* for fermentation. To assimilate the sucrose present in molasses, the SUC2 gene from *Saccharomyces cerevisiae* was expressed in *Yarrowia lipolytica* for sucrose utilization. Then, fermentation was carried out to obtain a yield of 52–114 g/L. *Candida magnoliae* is an osmophilic yeast and has an ability to assimilate sucrose, glucose, and fructose using the pentose phosphate pathway. Sucrose in the presence of invertase enzyme produces glucose and fructose. In the presence of hexokinase, glucose and fructose form G-6P (glucose-6-phosphate) and F-6P (fructose-6-phosphate), respectively. G-6P and F-6P in the presence of G-6P isomerase and Transketolase form erythrose-4-Phosphate as an intermediate. Erythrose-4-Phosphate is dephosphorylated by an enzyme, erythrose-4-P phosphatase, yielding erythrose, which is then reduced by an erythrose reductase (ER) into erythritol [17]. Thus, *Candida magnoliae* serves as an essential candidate for utilizing molasses as a substrate while not requiring any gene to be introduced for sucrose utilization [18].

Concomitantly, optimizing the media components and process condition is obligatory for the fermentative production of such metabolites as erythritol. Any fermentative production involves a set of several process and compositional factors. It is crucial to screen out the most significant factors among them. Following that, systematic optimization is necessary for those factors. The one-factor-at-a-time optimization method is time-intensive, and it only considers linear relationships and does not quantify or showcase the synergistic effects among the factors. For instance, Ghezelbash et al. [19] studied the erythritol production by varying concentrations of glucose, a nitrogen source, and pH one-factor-at-a-time, but the combined effects of these factors could not be known. Hence, obtaining a higher accuracy of the optimum value is difficult. Statistical methods such as response surface methodology (RSM) are more favorable as they consider the quadratic relationship between the factors, and the interaction among the factors can be calculated. Several statistical optimization tools, such as response surface modeling and the genetic algorithm, have been practiced in other studies [20–23]. Fed-batch fermentation to produce erythritol from glucose as a carbon source was optimized by Ryu et al. [24]. However, it was not a statistical media optimization. The systematic optimization of the microbial production of erythritol is limited. Therefore, the study optimizes the media components to produce erythritol using *Candida magnoliae*. Even though few studies have reported the production of erythritol using *Candida magnoliae* [19,24–26], the substrate used for the fermentation was pure glucose, which increases the production cost. In this study, we aimed to minimize the substrate cost by using molasses, which gave the erythritol a yield of 99.54 g/L, which is higher than the erythritol yield of 20.87 g/L obtained using pure glucose as substrate by Ghezelbash et al. [19]. The significant factors (media components) were initially screened out from the Plackett–Burman design. The media optimization was performed through RSM coupled with numerical optimization. Further, the purity of the erythritol produced at the optimized condition was characterized.

## 2. Materials and Methods

### 2.1. Materials

All reagents of analytical grade were purchased from HiMedia (Maharashtra, India). The yeast culture of *Candida magnoliae 3470* was procured from NCIM (Pune, India). Black-strap molasses was procured from Dhampure Specialty Sugars Ltd. (Delhi, India). The proximate composition of the molasses, such as moisture, crude protein, crude fat, reducing and non-reducing sugar, and ash content, was performed according to the protocol of AOAC International [27].

## 2.2. Culture Conditions for Measuring the Growth Curve

Liquid fermentation media (LFM) consisted of molasses (250 g·L$^{-1}$), yeast extract (10 g·L$^{-1}$), KH$_2$PO$_4$ (5 g·L$^{-1}$), and MgSO$_4$·7H$_2$O (0.25 g·L$^{-1}$). A loop-full of culture from freshly prepared slants of *Candida magnoliae 3470* was inoculated into 10 mL LFM in test tubes and incubated at 28 °C, 210 rpm, for 48 h. Two milliliters of this seed culture was aseptically transferred to 23 mL LFM, and the fermentation broth was incubated at 28 °C, 210 rpm for 168 h. The broth samples were withdrawn at successive intervals and analyzed for optical density (OD) at 600 nm. The dry cell weight (DCW), plate count, and change in pH of *C. magnoliae* were estimated from a corresponding standard curve. The erythritol yield was estimated using the erythrose reductase enzyme assay.

## 2.3. Preparation of Cell Extracts

*C. magnoliae* was grown in liquid fermentation media for 96 h, and the cell pellet was obtained by centrifugation at 10,000× *g* for 10 min. After washing the cells twice with 50 mM phosphate buffer (pH 6), 2 g of wet cells was suspended in 8 mL buffer (50 mM phosphate buffer pH 6, 10 mM MgCl$_2$) for 30 min. The cell suspension was homogenized by an ultrasonicator at 180 W, 70% duty cycle for 30 min with intermittent cooling to prevent denaturation of proteins. Enzyme extract was centrifuged at 10,000× *g* for 30 min at 4 °C. The supernatant was analyzed for protein content by Lowry's test using BSA (bovine serum albumin) as a standard to obtain 74.11 mg/mL of protein [28].

## 2.4. Erythrose Reductase Assay

The activity of erythrose reductase (ER) was quantified by the rate of NAD (nicotinamide adenine dinucleotide) consumption at 50 °C as determined spectrophotometrically at 340 nm. The ER assay mixture for oxidation contained 50 mM potassium phosphate buffer (pH 8), 50 μM NAD, 50 mM erythritol, and 0.1 mL of enzyme solution. One unit of ER activity represents 1 μmol of NAD consumed per minute from 1 mg of protein in the extract. The protocol for ER assay was validated using LC-MS by comparing the area under the curve of erythritol present in 0.5 ppm molasses fermentation broth and the same from 1 ppm pure erythritol solution.

## 2.5. Fermentative Production of Erythritol

A set of 11 variables was identified from the literature, which is reported to influence the fermentative production of erythritol. These are the concentration of molasses (carbon source), yeast extract (nitrogen source), KH$_2$PO$_4$ (phosphorus source), MgSO$_4$ (cofactor), and CaCO$_3$ (for pH stabilization), pH of the media, the volume of inoculum, temperature, time, agitation, and media volume. The Plackett–Burman design was employed to screen out the most influential factors among those 11 variables. In addition to erythritol yield, the dry cell weight and change in pH during fermentation were measured as two other responses to the Plackett–Burman design (Table 1).

The actual values of the independent variables ($X_i$) were converted into corresponding dimensionless coded values ($x_i$) using Equation (1), where $X_{max}$ and $X_{min}$ are the maximum and minimum values of $X_i$, respectively.

$$x_i = \frac{X_i - \frac{X_{max} + X_{min}}{2}}{X_{max} - \frac{X_{max} + X_{min}}{2}} \tag{1}$$

In this Plackett–Burman experimental design, each variable was varied at two levels (−1 and +1 in coded values), leading to 12 experimental runs. One hundred milliliters of fermentation media were prepared for each run with the desired composition and incubated at the set condition, as detailed in Table 1. A linear equation comprising all the factors was

developed from the least square error concept. The relative influence of each independent factor ($x_i$) on the response ($Y_i$) was calculated using Equation (2).

$$E = 2 \times \left( \sum Y_{i+} - \sum Y_{i-} \right) / n \tag{2}$$

where $E$ represents the influence of the factor, and $Y_{i+}$ and $Y_{i-}$ are the response values obtained for the factor $x_i$ varied at high (+1) and low (−1) levels, respectively, and $n$ represents the number of trials. From the relative contribution (%) values in the Pareto chart, a set of top three factors influencing the erythritol yield ($Y_1$, g·L$^{-1}$) was screened out (Table 1). These three variables were taken forward for optimization using response surface methodology (RSM).

**Table 1.** The experimental run of the Plackett–Burman design to screen out the most influential variables affecting the responses.

| Run | Independent Variables (Coded Value) | | | | | | | | | | | Response | | |
|---|---|---|---|---|---|---|---|---|---|---|---|---|---|---|
| | $X_1$ | $X_2$ | $X_3$ | $X_4$ | $X_5$ | $X_6$ | $X_7$ | $X_8$ | $X_9$ | $X_{10}$ | $X_{11}$ | $Y_1$ | $Y_2$ | $Y_3$ |
| | g·L$^{-1}$ | g·L$^{-1}$ | g·L$^{-1}$ | g·L$^{-1}$ | - | mL | °C | h | rpm | mL | mg·L$^{-1}$ | g·L$^{-1}$ | mg | - |
| 1 | 300 | 12 | 2 | 0.5 | 7 | 3 | 25 | 48 | 180 | 30 | 80 | 37.2 ± 1.8 | 350 ± 3 | 4.6 ± 0.0 |
| 2 | 200 | 12 | 5 | 0.1 | 7 | 3 | 30 | 48 | 180 | 20 | 120 | 32.6 ± 1.5 | 451 ± 4 | 4.3 ± 0.1 |
| 3 | 300 | 9 | 5 | 0.5 | 4 | 3 | 30 | 96 | 180 | 20 | 80 | 105.7 ± 4.3 | 482 ± 4 | 3.0 ± 0.1 |
| 4 | 200 | 12 | 2 | 0.5 | 7 | 1 | 30 | 96 | 240 | 20 | 80 | 22.5 ± 1.5 | 733 ± 5 | 5.2 ± 0.0 |
| 5 | 200 | 9 | 5 | 0.1 | 7 | 3 | 25 | 96 | 240 | 30 | 80 | 35.1 ± 1.6 | 934 ± 8 | 4.2 ± 0.0 |
| 6 | 200 | 9 | 2 | 0.5 | 4 | 3 | 30 | 48 | 240 | 30 | 120 | 36.5 ± 1.7 | 592 ± 4 | 3.1 ± 0.0 |
| 7 | 300 | 9 | 2 | 0.1 | 7 | 1 | 30 | 96 | 180 | 30 | 120 | 96.6 ± 4.9 | 174 ± 2 | 4.6 ± 0.1 |
| 8 | 300 | 12 | 2 | 0.1 | 4 | 3 | 25 | 96 | 240 | 20 | 120 | 36.1 ± 1.8 | 664 ± 3 | 3.7 ± 0.1 |
| 9 | 300 | 12 | 5 | 0.1 | 4 | 1 | 30 | 48 | 240 | 30 | 80 | 94.5 ± 4.7 | 613 ± 3 | 3.9 ± 0.1 |
| 10 | 200 | 12 | 5 | 0.5 | 4 | 1 | 25 | 96 | 180 | 30 | 120 | 23.8 ± 1.6 | 742 ± 4 | 3.4 ± 0.0 |
| 11 | 300 | 9 | 5 | 0.5 | 7 | 1 | 25 | 48 | 240 | 20 | 120 | 125.9 ± 4.9 | 585 ± 3 | 4.7 ± 0.1 |
| 12 | 200 | 9 | 2 | 0.1 | 4 | 1 | 25 | 48 | 180 | 20 | 80 | 33.5 ± 1.5 | 464 ± 3 | 3.0 ± 0.0 |
| t-Stat (Y1) | 52.02 | −31.13 | 25.88 | 3.89 | 3.31 | −18.94 | 16.08 | −6.75 | 3.5 | −5.45 | 3.83 | | | |
| %Cont (Y1) | 53.15 | 19.04 | 13.16 | 0.30 | 0.22 | 7.04 | 5.08 | 0.89 | 0.24 | 0.58 | 0.29 | | | |
| Rank (Y1) | I | II | III | VIII | XI | IV | V | VI | X | VII | IX | | | |

$X_1$ = molasses concentration, $X_2$ = yeast extract concentration, $X_3$ = KH$_2$PO$_4$ concentration, $X_4$ = MgSO$_4$ concentration, $X_5$ = pH, $X_6$ = inoculum volume, $X_7$ = temperature, $X_8$ = time, $X_9$ = agitation, $X_{10}$ = media volume, $X_{11}$ = CaCO$_3$, respectively. Y1, Y2, Y3 are erythritol yield, DCW, and ΔpH, respectively. %cont is the percentage contribution for Y1.

### 2.6. Response Surface Methodology

A rotatable central composite design (RCCD) was employed on the three screened-out factors: molasses ($X_1$ g·L$^{-1}$), yeast extract ($X_2$ g·L$^{-1}$), and KH$_2$PO$_4$ ($X_3$ g·L$^{-1}$). The domain of the variables was selected from the literary information. The RCCD matrix resulted in 20 experimental runs with $2^3$ factorial runs (±1 level in coded values), $2 \times 3$ axial runs (±α level in coded values), and 6 repeated runs at the center point (0 levels in coded values) (Table 2).

The response measured was the erythritol yield ($Y_1$ g·L$^{-1}$) in the medium after fermentation. A quadratic polynomial model was developed for erythritol yield ($Y_1$ g·L$^{-1}$) as a function of three independent variables, viz., $x_1$, $x_2$, and $x_3$, as described in Equation (3).

$$Y_1 = \beta_o + \beta_1 x_1 + \beta_2 x_2 + \beta_3 x_3 + \beta_4 x_1 x_2 + \beta_5 x_1 x_3 + \beta_6 x_2 x_3 + \beta_7 x_1{}^2 + \beta_8 x_2{}^2 + \beta_9 x_3{}^2 \tag{3}$$

where $\beta_0$ to $\beta_9$ are the regression coefficients generated from minimizing the sum of the square of errors using Equation (4).

$$\beta = [D'D]^{-1} D' y_a \tag{4}$$

In Equation (4), $[\beta]_{10 \times 1}$ is the matrix consisting of $\beta_0$ to $\beta_9$; $[D]_{20 \times 10}$ is the matrix composed of 10 parameters from Equation (3) and the corresponding values for 20 experimental runs. The matrix $[y_a]_{20 \times 1}$ is the actual set of response values for 20 experimental runs. The adequacy of model fitting was checked by analysis of variance (ANOVA) data. The $R^2$ value was used to know the overall predictive capability of the model. The statistical significance of the fit of the polynomial model equation was checked by the variance test (F-test) with a confidence interval of 95% of the mean. The significance of the regression coefficient was tested by $p$-value (probability of accepting null hypothesis). The response surface model and contour plots were developed for the erythritol yield as a function of the two variables between $x_1$, $x_2$, and $x_3$.

**Table 2.** Rotatable central composite design, corresponding erythritol yield, and predicted results for the polynomial model.

| Run | Independent Variable (Coded Value) | | | Erythritol Yield | | Error |
|---|---|---|---|---|---|---|
| | Molasses | Yeast Extract | KH$_2$PO$_4$ | Actual | Predicted | |
| | g·L$^{-1}$ | g·L$^{-1}$ | g·L$^{-1}$ | g·L$^{-1}$ | g·L$^{-1}$ | % |
| 1 | 200 (−1) | 9 (−1) | 2 (−1) | 35.1 ± 1.8 | 33.5 | +4.5 |
| 2 | 200 (−1) | 9 (−1) | 5 (+1) | 96.6 ± 3.9 | 88.1 | +8.7 |
| 3 | 200 (−1) | 12 (+1) | 2 (−1) | 63.1 ± 2.3 | 60.0 | +4.9 |
| 4 | 200 (−1) | 12 (+1) | 5 (+1) | 72.5 ± 2.7 | 76.1 | −4.9 |
| 5 | 300 (+1) | 9 (−1) | 2 (−1) | 59.2 ± 2.5 | 51.6 | +12.7 |
| 6 | 300 (+1) | 9 (−1) | 5 (+1) | 89.4 ± 3.5 | 86.0 | +3.8 |
| 7 | 300 (+1) | 12 (+1) | 2 (−1) | 62.3 ± 2.4 | 64.3 | −3.2 |
| 8 | 300 (+1) | 12 (+1) | 5 (+1) | 65.2 ± 2.9 | 60.2 | +7.6 |
| 9 | 166 (−1.68) | 10.5 (0) | 3.5 (0) | 28.2 ± 1.7 | 30.9 | −9.6 |
| 10 | 334 (+1.68) | 10.5 (0) | 3.5 (0) | 68.7 ± 3.1 | 73.4 | −6.8 |
| 11 | 250 (0) | 7.98 (−1.68) | 3.5 (0) | 57.8 ± 3.2 | 63.5 | −9.8 |
| 12 | 250 (0) | 13.02 (+1.68) | 3.5 (0) | 65.8 ± 3.6 | 64.1 | +2.6 |
| 13 | 250 (0) | 10.5 (0) | 0.98 (−1.68) | 80.8 ± 3.3 | 83.3 | −3.1 |
| 14 | 250 (0) | 10.5 (0) | 6.02 (+1.68) | 78.6 ± 3.1 | 85.2 | −8.4 |
| 15 | 250 (0) | 10.5 (0) | 3.5 (0) | 89.4 ± 3.6 | 96.6 | −8.0 |
| 16 | 250 (0) | 10.5 (0) | 3.5 (0) | 105.6 ± 4.6 | 96.6 | +8.5 |
| 17 | 250 (0) | 10.5 (0) | 3.5 (0) | 104.2 ± 4.5 | 96.6 | +7.3 |
| 18 | 250 (0) | 10.5 (0) | 3.5 (0) | 89.4 ± 4.8 | 96.6 | −8.0 |
| 19 | 250 (0) | 10.5 (0) | 3.5 (0) | 97.1 ± 5.0 | 96.6 | +0.5 |
| 20 | 250 (0) | 10.5 (0) | 3.5 (0) | 95.0 ± 4.9 | 96.6 | −1.6 |

Error = [(Actual yield − Predicted yield)/Actual yield] × 100%.

### 2.7. Numerical Optimization

The numerical optimization was applied to find the optimum combination of $x_1$, $x_2$, and $x_3$ targeting a maximum erythritol yield ($Y_1$ g·L$^{-1}$). It was targeted to maximize the value of the overall desirability function ($d_1$) according to Equation (5).

$$d_1 = \left( \frac{Y_1 - L_1}{U_1 - L_1} \right)^w \tag{5}$$

where $d_1$ is the desirability index for the response (erythritol yield, $Y_1$); the value of $d_1$ ranges between 0 and 1, and a magnitude nearer to 1 represents the most desirable ones. The weightage ($w$) represents the nature of the curve followed by $d_1$ between 0 to 1. It is taken as a linear pathway ($w$ = 1). $L_1$ and $U_1$ are the lower and upper limits of the $Y_1$.

### 2.8. Analysis of Fermentation Broth

The erythritol concentration from the fermentation broth was determined using erythrose reductase enzyme assay and validated using LC-MS as per the protocol detailed by Savergave et al. [18]. For instance, 0.5 mM NAD was the cofactor, and the stock substrate solution was 50 mM of erythritol. *Candida magnoliae* was the corresponding microbial strain.

The mixture was incubated at 50 °C for 5 min at pH 8. The reading was taken using a microplate reader. The concentrations of glucose, sucrose, mannitol, erythritol, glycerol, and other co-metabolites were determined using liquid chromatography–mass spectrometry (LC-MS) (Shimadzu 8040 Triple Quadra pole, Shimadzu, Kyoto, Japan) equipped with a C18 column (4.6 × 250 mm × 5 μm). The mobile phase was composed of methanol and 0.1% formic acid (7:3 *v/v*) at a flow rate of 1 mL·min$^{-1}$ with a total run time of 6 min. The quantification was performed by a standard external technique using the peak area of reference compounds. The standard erythritol of 1 μg·mL$^{-1}$ solution showed a 1,347,578 nm$^2$ area under the curve in LC-MS.

### 2.9. Purification of Fermentation Broth

The purification of fermentation broth was derived by Savergave [18] with some modifications. The product (erythritol) was purified from 500 mL fermentation broth. The broth was centrifuged at 8000× *g* for 10 min, and the supernatant was treated with 1% activated charcoal at 90 °C for 20 min under gentle agitation. The activated carbon was removed using Whatman (#1) filter paper. The clear solution obtained was evaporated in a rotary evaporator at 70 °C under a vacuum to concentrate erythritol to around 200 g·L$^{-1}$. The concentrated solution was then allowed to cool to 20 °C under gentle agitation and seeded with a trace amount of erythritol to initiate crystallization. The solution was then incubated at 4 °C overnight. Brittle white erythritol crystals formed were collected by filtration, washed twice with cold distilled water, and dried at 50 °C for two hours in a hot air oven. Further, the purified erythritol crystals were characterized by LC-MS by comparison with the mass spectra of standard or reference erythritol. The injection volume of the molasses sample was 10 μL comprising 5% broth, and it was compared with pure erythritol solution (1 g/L). The concentration of erythritol present in the molasses was calculated by comparing the ratio between the area under the curve of standard erythritol and molasses broth, respectively.

### 2.10. Statistical Analysis

The fermentation experiments were conducted in duplicate, and the corresponding erythritol yield was assayed in triplicate. In the case of the Plackett–Burman design, the responses were analyzed in triplicate. The screening of the factors using Plackett–Burman design, the response surface modeling, and numerical optimization were attempted in Design-Expert software (Version 8.0.2.0, Stat-Ease Inc., Minneapolis, MN, USA). The significance test and analysis of variance (ANOVA) were accompanied by Tukey's HSD test conducted in SPSS software (IBM SPSS Statistics 16, New York, NY, USA).

## 3. Results and Discussion

### 3.1. Growth Curve of Candida magnoliae

The biomass generation for *Candida magnoliae* followed a typical growth pattern for a batch culture. The growth curve of *Candida magnoliae* represents an initial lag phase of 5 h, occupied with the acclimatization with the substrate. The exponential phase starts after 5 h from incubation, and a rapid increase in the absorbance accompanies it until 20 h. The stationary phase was stable from 25 h, and the corresponding death phase started at 30 h. Irrespective of the substrate and strain, the trends for the growth curve were similar. A linear relationship was obtained between the measured absorbance (OD$_{600nm}$) and the dry cell weight (DCW, mg) (Equation (6)). An exponential relationship connected the cell count (N cfu/mL) to OD by an exponential relationship (Equation (7)).

$$\text{DCW} = 0.99 + 1.757 \times \text{OD} \tag{6}$$

$$\text{N} = 1.69 \times 10^7 [\exp(\text{OD}/0.169)] - 1.947 \times 10^8 \tag{7}$$

The adjusted R$^2$ for Equations (6) and (7) were 0.92 and 0.99, respectively. This indicated the adequacy of model fitting.

### 3.2. Screening of Significant Factors Using Plackett–Burman Design

The Plackett–Burman design searched for significant factors that substantially affect erythritol production, and the design and analysis are summarized in Table 1. Within the domain of Plackett–Burman design, the erythritol yield ($Y_1$) varied between 22.5 between 125.9 g·L$^{-1}$. The DCW ranged from 170 to 930 mg, whereas $\Delta$pH varied between 3.0 and 5.3. A set of linear equations has evolved to describe the changes in three responses as a function of independent variables ($x_1$ to $x_{11}$). The equation for the responses in terms of coded value is presented in Equation (8).

$$Y_1 = 56.68 + 26.01x_1 - 15.57x_2 + 12.94x_3$$
$$Y_2 = 564 - 87x_1 + 69x_3 + 120x_{10} \tag{8}$$
$$Y_3 = 4.01 + 0.21x_2 + 0.657x_5 - 0.15x_6$$

where $x_1$–$x_{11}$ represent the coded values ($-1$ to $+1$) for the independent variables $X_1$–$X_2$ in Table 1. Here, $-1$ corresponds to the lower limit and $+1$ corresponds to the upper limit of the domain. The adjusted R$^2$ for the linear equations obtained for $Y_1$, $Y_2$, and $Y_3$ were 0.85, 0.75, and 0.91, respectively; the respective F-values were 15.54, 7.72, and 28.4. This reflects that the changes in the response values are not due to noise; rather, these are influenced by the factors. The significant factors ($p$-value < 0.05) were screened out for each response. Interestingly, for each equation, only three independent variables were significant. For instance, $x_1$, $x_2$, and $x_3$ influenced $Y_1$ significantly; $Y_2$ was affected most by $x_1$, $x_3$, and $x_{10}$; and the contribution of $x_2$, $x_5$, and $x_6$ was greatest for $Y_3$. The coefficient in the coded form equation (Equation (8)) shows that molasses ($x_1$) was the most significant factor influencing the erythritol formation ($Y_1$) positively. The positive influence signifies that with an increase in molasses concentration (factor), the erythritol yield (response) is improved or increased. In addition, increasing the concentration of KH$_2$PO$_4$ ($x_3$) increased the $Y_1$ significantly, whereas yeast extract ($x_2$) negatively influenced erythritol production. On the contrary, it was found that molasses concentration ($x_1$) negatively contributed to dry cell weight ($Y_2$), whereas the more KH$_2$PO$_4$ ($x_3$) and media volume ($x_{10}$) were used, the greater the dry cell weight obtained. It was also found that yeast extract ($x_2$) and inoculum volume ($x_6$) significantly influenced the change in pH as per Equation (8). Therefore, the concentrations of molasses ($x_1$), yeast extract ($x_2$), and KH$_2$PO$_4$ ($x_3$) were taken forward in the next step for the optimization of erythritol yield ($Y_1$).

### 3.3. Response Surface Modeling of Erythritol Yield

The sources for carbon (molasses, $x_1$), nitrogen (yeast extract, $x_2$), and phosphate (KH$_2$PO$_4$, $x_3$) were the critical medium components for erythritol production ($Y_1$). These three factors significantly influenced erythritol yield ($Y_1$) (Table 2). The yield ranged between 28.2 and 105.6 g·L$^{-1}$. Increasing molasses (carbon source) led to a higher product yield. For instance, at 9 g·L$^{-1}$ yeast extract and 2 g·L$^{-1}$ KH$_2$PO$_4$, the erythritol yield increased from 35.1 to 59.2 g·L$^{-1}$ when the molasses concentration was increased from 200 to 300 g·L$^{-1}$ (Table 2). Molasses contains different carbon and nitrogen sources such as sucrose, thiamine, and so on, which are easily assimilated, supporting the cell growth of osmotolerant microbes. The components of molasses vary greatly. Molasses is complex, and it contains mainly sucrose besides other components. The proximate composition of the molasses used in this study is summarized in Table 3. The molasses' reducing and non-reducing sugar contents are 181 and 352 g·L$^{-1}$, respectively. The cumulative concentration of nitrogenous compounds (in terms of crude protein) was 3.5 g·L$^{-1}$, whereas fat content was 4.2 g·L$^{-1}$. The molasses was black and contained 22.5% moisture with 11.5% ash.

On the other hand, at 9 g·L$^{-1}$ yeast extract and 5 g·L$^{-1}$ KH$_2$PO$_4$, the yield reduced from 96.6 to 89.4 g·L$^{-1}$ when the molasses concentration was increased from 200 to 300 g·L$^{-1}$ (Table 2). The osmotic stress might be responsible for this trend in erythritol yield [26]. The osmotic pressure of the medium increased from 0.988 to1.482 kPa when the molasses concentration changed from 200 to 300 g·L$^{-1}$. The dry cell weight of the

biomass recovered from 1 L medium decreased from 20.23 to 18.36 g. The reduction in cell biomass might be due to the unavailability of soluble oxygen in the medium. In addition, the C:N ratio might play a role in diverting the balance between biomass production and polyol formation [29].

**Table 3.** Constituents of blackstrap cane molasses used in the study.

| Constituents (Unit) | Quantity |
|---|---|
| Moisture (g/100 g) | $22.5 \pm 4.3$ |
| Total sugar (g/100 g) | $53.3 \pm 5.6$ |
| Reducing sugar (g/100 g) | $18.1 \pm 2.5$ |
| Non-reducing sugar (g/100 g) | $35.2 \pm 3.2$ |
| Crude protein (g/100 g) | $0.35 \pm 0.1$ |
| Crude fat (g/100 g) | $0.42 \pm 0.1$ |
| Ash (g/100 g) | $11.5 \pm 4.0$ |
| Brix value (°Bx) | $78.1 \pm 0.2$ |
| pH (-) | $5.9 \pm 0.1$ |

With 200 $g \cdot L^{-1}$ molasses and 2 $g \cdot L^{-1}$ $KH_2PO_4$ mixed in the fermentation medium, the erythritol yield increased from 35.1 to 63.1 $g \cdot L^{-1}$ when the yeast extract concentration increased from 9 to 12 $g \cdot L^{-1}$, respectively. At 250 $g \cdot L^{-1}$ molasses and 3.5 $g \cdot L^{-1}$ $KH_2PO_4$, the yield was elevated from 57.8 to 65.8 $g \cdot L^{-1}$ for 7.98 to 13.02 $g \cdot L^{-1}$ yeast extract, respectively. Yeast extract is commonly used as the nitrogen source for the fermentative production of sugars such as erythritol. It is a source of thiamine required to produce erythritol [7]. The yeast powder contained ($w/w$) protein (30%), fat (0.42%), sodium chloride (0.67%), ash (12.18%), and total volatile nitrogen (9.2%) with a moisture of 4.72% and pH of 6.29. Molasses contained a certain amount of nitrogen compounds (3.5 $g \cdot L^{-1}$), as indicated in Table 3. Both yeast extract and molasses provide organic nitrogen; thus, an increase in either component led to a higher erythritol yield in the medium [17].

Moreover, an increase in $KH_2PO_4$ concentration in the medium significantly enhanced the erythritol yield. Keeping the molasses (200 $g \cdot L^{-1}$) and yeast extract (9 $g \cdot L^{-1}$) fixed, the product yield showed an elevation from 35.1 to 96.6 $g \cdot L^{-1}$ when the phosphate concentration increased from 2 to 5 $g \cdot L^{-1}$. However, after reaching an optimum, a reverse trend was found. For instance, $Y_1$ was 80.8 and 78.6 $g \cdot L^{-1}$ when the $KH_2PO_4$ concentration increased from 0.98 to 6.02 $g \cdot L^{-1}$ at 250 $g \cdot L^{-1}$ molasses and 10.5 $g \cdot L^{-1}$ yeast extract. For the growth of *Candida* species, the role of inorganic phosphate is indispensable. In addition, a higher concentration of KH2PO4 led to a higher osmotic stress surrounding the *Candida magnoliae*. This eventually might promote the cellular erythritol production so that osmotic stress inside the cell is balanced [10,30].

The quadratic polynomial model fitted best to the experimental data of erythritol yield ($Y_1$) as a function of molasses ($x_1$), yeast extract ($x_2$), and $KH_2PO_4$ concentration ($x_3$). The F-value for the quadratic model was 10.62 compared to the same 1.67 and 0.67 for the linear and factorial interaction model, respectively. The cubic model showed an F-value of 0.59. Therefore, the quadratic polynomial model was taken forward to visualize the square and interaction effect between $x_1$, $x_2$, and $x_3$ on the response. The corresponding lack of fit *p*-value was 0.4539 (insignificant). Lack of fit signifies that the influence of noise variables on the response. Statistically, an insignificant lack of fit is expected for a model to be fit. Additionally, an insignificant lack of fit refers that any change in response is well connected to the variation in the controllable factors such as $x_1$, $x_2$, and $x_3$ (Table 4).

The lack of fit is not significant relative to the pure error. The model F-value of 10.62 implied that the model was significant at $p < 0.01$, negating any noise within the dataset. A difference of 0.08 between the regression coefficient ($R^2 = 0.92$) and adjusted $R^2$ (0.84) depicts the model adequacy. The predicted response values are also reflected in Table 3. The percentage error between the actual and predicted values is below 10%, except for the outlier at one condition (run: 5, showing an error of 12.7%). This signifies that except that

outlier, the deviation in the responses from the fitted line is within 10%. A low value of the coefficient of variation of 12.40% indicates a very high degree of precision and good reliability of the experimental values. The coefficient of variation dictates the relative size of the standard deviation in comparison to the fitted value including all the outliers. A coefficient of variation of 12.40% is well accepted from a statistical point of view. The adequacy precision, which measures the signal-to-noise ratio, shows a value of 9.9, whereas it is desirable to have this value > 4. This quadratic polynomial model navigates the design space when presented graphically. The points accommodate themselves along the diagonal, indicating the high level of statistical significance of the model.

**Table 4.** The analysis of variance (ANOVA) data for the response surface model developed for erythritol yield ($Y_1$ g·L$^{-1}$).

| Parameter | Coefficient ± 95% CI | *p*-Value |
|:---:|:---:|:---:|
| Constant | 96.58 ± 8.41 | 0.0005 |
| $x_1$ | 12.63 ± 5.61 | 0.0005 |
| $x_2$ | 0.18 ± 2.52 | 0.9433 |
| $x_3$ | 0.55 ± 2.51 | 0.8301 |
| $x_1 x_2$ | −9.61 ± 3.29 | 0.0153 |
| $x_1 x_3$ | −5.05 ± 3.66 | 0.1004 |
| $x_2 x_3$ | −3.44 ± 3.28 | 0.3200 |
| $x_1 x_1$ | −15.68 ± 5.47 | <0.0001 |
| $x_2 x_2$ | −11.57 ± 5.46 | 0.0008 |
| $x_3 x_3$ | −4.34 ± 2.05 | 0.0093 |
| F-value | 10.52 | - |
| *p*-value (model) | - | 0.0002 |
| *p*-value (lack of fit) | - | 0.4539 |
| R$^2$ | 0.92 | - |
| Adj R$^2$ | 0.84 | - |

The linear terms of $x_1$, $x_2$, and $x_3$ in the polynomial equation positively influence the response, which implies that on increasing the concentration of either of these three components, the yield of erythritol also increases significantly. The major contributor to increasing the erythritol yield is molasses concentration, followed by KH$_2$PO$_4$ and yeast extract. The interaction between molasses and yeast extract concentration ($x_1 x_2$) is significant at $p < 0.05$, whereas other interaction terms, such as $x_1 x_3$ and $x_2 x_3$, do not statistically ($p > 0.1$) influence the response ($Y_1$). All of the three-square terms ($x_1 x_1$, $x_2 x_2$, and $x_3 x_3$) showed significant ($p < 0.05$) negative coefficients in the polynomial model. The negative square terms support the parabolic nature of the curve, corroborating the notion that, initially, the yield will increase with an increase in either $x_1$, $x_2$, or $x_3$. However, the yield will be compromised beyond a particular concentration, and the shape of the curve will be reversed. Figure 1 depicts the changes in erythritol yield with respect to a simultaneous change in molasses and yeast extract concentration. This contour plots are drawn at a fixed phosphate concertation of 3.5 g/L. The contour plot showing the interaction between $x_1$ and $x_2$ on the response follows this parabolic trend. When the concentration of molasses and yeast extract was 200 and 9 g·L$^{-1}$, the erythritol yield was 55.6 g·L$^{-1}$, whereas when the concentration was increased to 250 and 11 g·L$^{-1}$, the erythritol yield increased to 90 g·L$^{-1}$. However, the yield of erythritol declined to 81.9 g·L$^{-1}$ when the concentration was further increased to 280 and 11.3 g·L$^{-1}$. A similar influence of glucose concentration on erythritol yield was observed by Park et al. [31]. In another study, it was seen that glycerol started to appear in the broth as a by-product instead of erythritol production. This decline can be attributed to an increase in the osmolarity of the fermentation media, which might reduce the oxygen transfer rate towards the cells, which is crucial for erythritol formation [26].

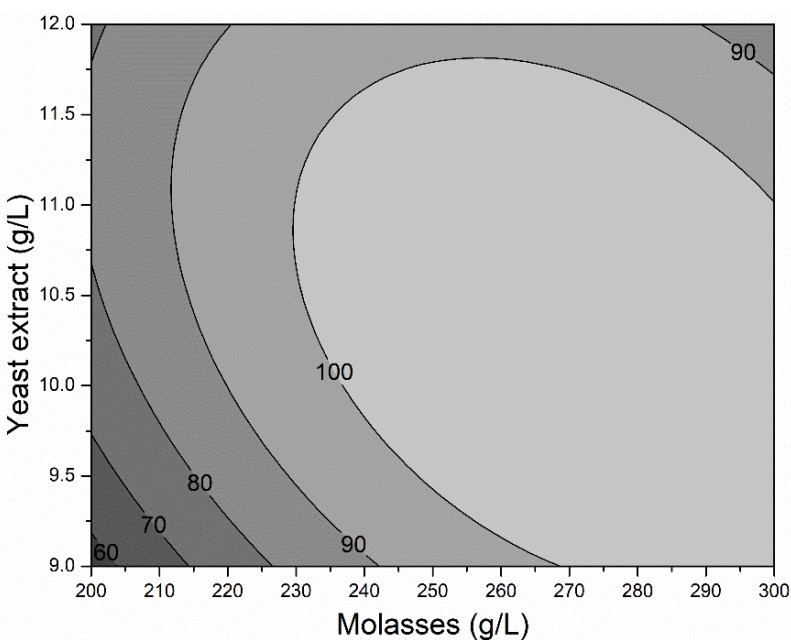

**Figure 1.** The contour plots of erythritol yield in molasses and yeast extract concentration landscape at fixed phosphate concertation of 3.5 g/L.

### 3.4. Numerical Optimization of Fermentation Conditions

The contour plots generated from the fitted quadratic model showed the influence of the maximum two variables on the erythritol yield at a given condition. However, the optimization condition is expected to belong to any point within the domain. In turn, the optimization of fermentation conditions was performed by iterating the maximum desirability value within the selected three-dimensional domain of $x_1$, $x_2$, and $x_3$. The desirability value ($D$) is connected to $Y$, which is again the function of the independent variables. The desirability value ($D$) was maximized, leading to a maximum erythritol yield. The relative importance of the response was 5 out of 5. The maximum erythritol yield predicted was 99.54 g·L$^{-1}$ under the optimized condition of molasses, yeast extract, and $KH_2PO_4$ concentrations of 273.96, 10.25, and 3.28 g·L$^{-1}$, respectively. A high desirability value of 0.88 supported this optimized condition. The actual experimental yield of erythritol obtained was 98.89 g·L$^{-1}$. The error during validation was less than 1%, corroborating the model fitting accuracy. Similar results of erythritol yield have been reported in the literature. For instance, Hijosa-Valsera et al. [32] reported an erythritol yield of 106.4 g·L$^{-1}$ from 300 g·L$^{-1}$ molasses using the yeast *Moniliella pollinis*. In another study, a three-level factorial design with glycerol, urea, and NaCl as the three most contributing factors for erythritol yield was used. The predicted erythritol yield was 100.6 g·L$^{-1}$ using 214.5 g·L$^{-1}$ glycerol with 1.69 g·L$^{-1}$ urea and 37.2 g·L$^{-1}$ NaCl [33]. Savergave et al. [26] used a four-level factorial design for the optimization of erythritol, which included glucose 238 g·L$^{-1}$, yeast extract 9.2 g·L$^{-1}$, $KH_2PO_4$ 5.16 g·L$^{-1}$, and $MgSO_4$ 0.23 g·L$^{-1}$ as factors. They predicted the optimized erythritol yield of 87.8 g·L$^{-1}$. At the optimized condition, the erythritol yield was 0.625 g/g of dry cell weight. Deshpande et al. [34] obtained an erythritol yield of 0.38 g/g of dry cell weight when molasses was used as a feedstock by *Moniliella pollinis*. Rakicka et al. [35] employed a two-stage chemostat process with glycerol and reported an erythritol yield of 0.66 g/g of dry cell weight.

### 3.5. Purification and Characterization of Erythritol

The percentage purity of erythritol crystals obtained from the purification of broth was calculated by dividing the area under the curve of 1 ppm crystallized erythritol by 1 ppm of standard erythritol obtained through LC-MS data. The value obtained was operated under MRM mode as it is more sensitive and provided less matrix interference, which is

also mentioned by Chang and Yeh [33]. The area under the purified and standard erythritol crystals curve showed values of 2,337,593 and 2,486,912 nm$^2$, respectively (Figure 2).

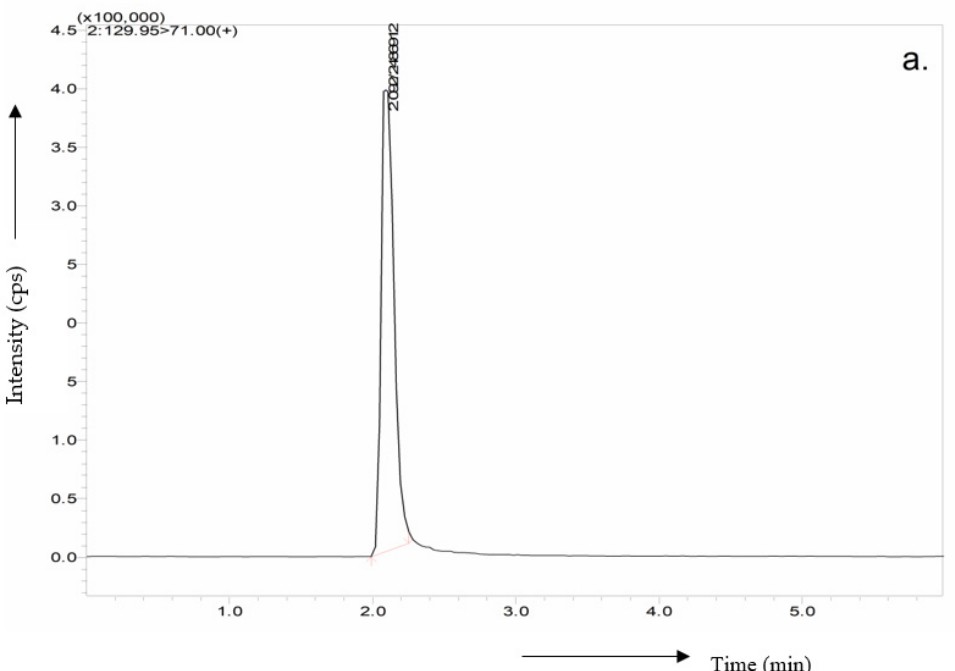

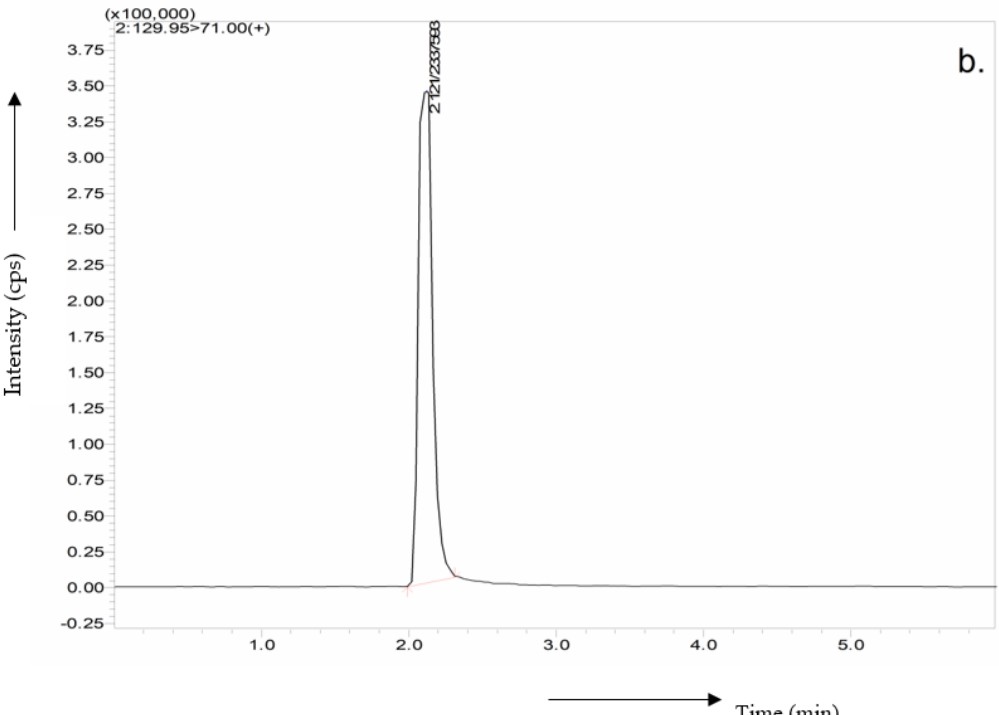

**Figure 2.** LC-MS spectra of (**a**) 1 mg/L purified erythritol and (**b**) 1 mg/L standard erythritol crystals.

This reflects that the obtained erythritol crystals possessed ~94% purity [36]. The LC-MS overlay chromatogram of the crystalline erythritol formed after fermentation revealed that glycerol was obtained as a by-product (Figure 3).

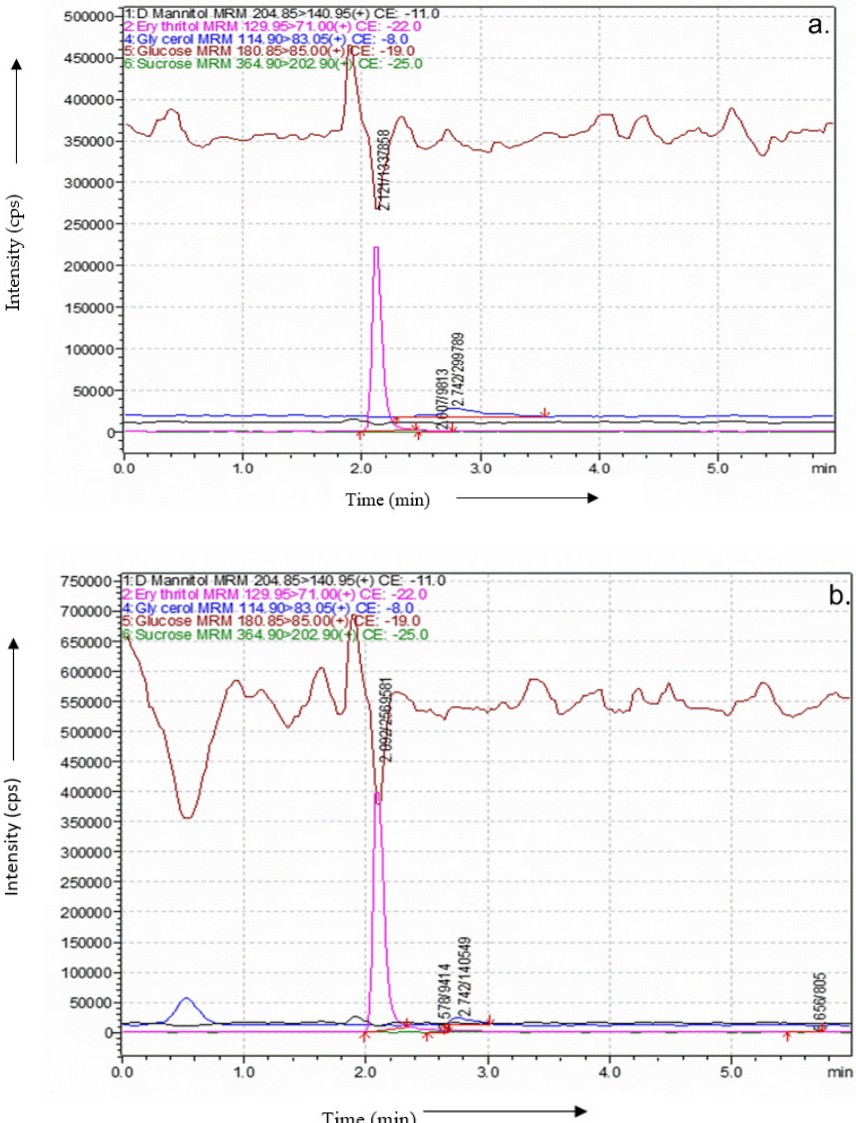

**Figure 3.** LC-MS overlay chromatogram with (**a**) 1 ppm standard erythritol and (**b**) crystalline erythritol obtained after fermentation.

A trace amount of mannitol was also detected. The glycerol concentration was 5.4% of the pure erythritol, whereas the sucrose concentration was 0.36%. A similar trend has been reported in the literature showcasing mannitol, glycerol, arabitol, lactic acid, acetic acid, and ethanol as different by-products during the fermentative production of erythritol [26,37,38]. Surprisingly, the area under the curve of the erythritol crystal was greater than the commercial erythritol crystal. The MS spectra were obtained to detect and confirm different products and by-products obtained from fermentation broth in the positive ion mode (Figure 4).

The erythritol's mass spectra and a protonated peak with ($m/z$) 123.15 showed an additional adducted sodium ion sharp peak. Sodium added ions were detected as base peaks, and the molecular masses of the erythritol were determined by comparing [M + Na] + ions. Analysis in electrospray (ES) positive mode gave ($m/z$) 145.12, i.e., [M + 23]. The same trends were reported by Adeuya and Prince [39] when erythritol was trimethylsilylated and analyzed using MALDI-TOF-MS leading to an $m/z$ ratio of 145 Da due to the presence of sodium adducts. The sodium adducts were seen in other sugar alcohols such as xylitol and mannitol. Similar trends were studied by Yoon et al. [40], where erythritol was transglycosylated into maltosyl-erythritol by the bacteria *Bacillus stearothermophilus* where

the molecular masses of proton, sodium, and potassium adducts of maltosyl-erythritol using an LC-MS plot were found to be 447, 468, and 485 Da, respectively. This trend is very similar to the nitrate adduct formed with erythritol described by Forbes and Sisco [41], where the $m/z$ ratio of $NO_3^-$ is 61.987, while the $m/z$ of [erythritol + $NO_3$]$^-$ is 184.046. The additional peaks in the mass spectra denote the presence of minor compounds associated with the erythritol crystals involved in the purification and crystallization steps.

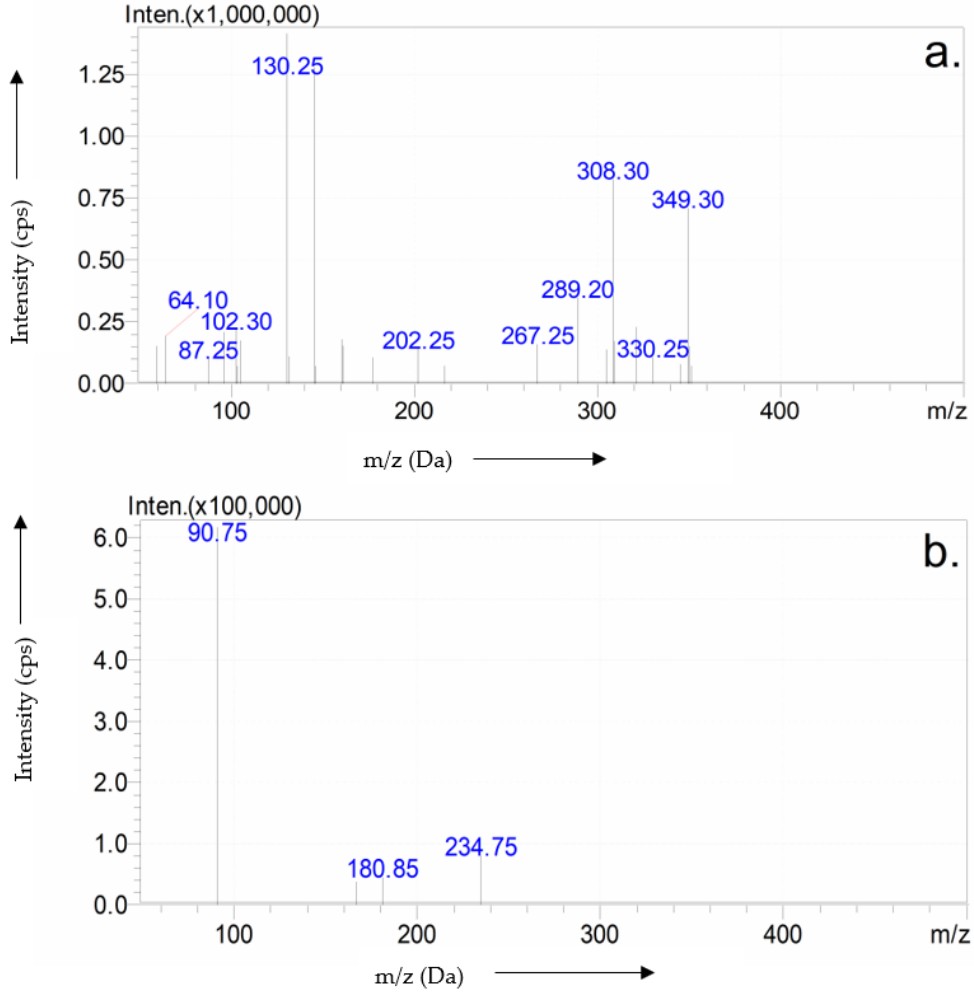

**Figure 4.** Mass spectra of (**a**) 1 ppm standard erythritol and (**b**) crystalline erythritol obtained after fermentation.

The erythritol yield of 99.54 g·L$^{-1}$ was optimized using response surface methodology using 273.96 g·L$^{-1}$ molasses, 10.25 g·L$^{-1}$ yeast extract, and 3.28 g·L$^{-1}$ KH$_2$PO$_4$ in the medium, which was comparable to or higher than the erythritol yield obtained using glucose fermentation by *Candida magnoliae*. Moreover, the study uses molasses, a by-product of the sugarcane industry, for the production of erythritol, bringing the process a step closer toward attaining sustainability as opposed to using commonly used substrates such as pure glucose/fructose, which increases the production cost. The erythritol yield was estimated by erythrose reductase enzyme assay, and the results obtained were validated using LC-MS. The erythritol crystals purified from the fermentation broth showed 94% purity. Further study should focus on identifying the crystal structure of the obtained erythritol. Improving the yield of erythritol by gene modification through site-specific mutagenesis of *Candida magnoliae* can be studied further. The influence of flocculants and the yeast grown in various molasses on the erythritol yield may be explored. Besides, scaling-up studies of fermentation media inside a bioreactor would be of great interest.

**Author Contributions:** Conceptualization, S.S. and S.C.; methodology, S.S.; software, S.S.; validation S.S. and S.C.; formal analysis, S.S.; investigation, S.S. and S.C.; resources, S.C.; data curation, S.S.; writing—original draft preparation, S.S.; writing—review and editing, S.C.; visualization, S.S. and S.C.; supervision, S.C.; project administration, S.C.; funding acquisition, S.C. All authors have read and agreed to the published version of the manuscript.

**Funding:** This research was funded by the Department of Biotechnology, Government of India under the grant number BT/HRD/01/007/2007 Vol II.

**Institutional Review Board Statement:** This research does not include studies or trials with human participants or animals. Hence, no formal consent is required.

**Informed Consent Statement:** Not applicable.

**Data Availability Statement:** Both the authors confirm that all the data generated from this research are provided within this article.

**Conflicts of Interest:** The authors declare no conflict of interest.

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
