# Peer review of "Fermentative Production of Erythritol from Cane Molasses Using Candida magnoliae: Media Optimization, Purification, and Characterization"

_sustainability, doi:10.3390/su141610342_

Round 1

Reviewer 1 Report

The current study explored the fermentative production of erythritol from cane molasses using Candida magnolia. The whole work is interesting and valuable. However, there are some points that should be improved in the revised version:

1. The Table 1 is too busy to be read or understand, which should be revised. 

2. There are only two references that related with the the fermentative production of erythritol by Candida magnolia, and one reference from 2000, the other from 2011. It is hard to believe that such few people did the related work. More reference should be added in the introduction and discussion part regarding to this topic. And compare to these previous work, what's new in the current study? The yield is improved, the purity is elevated or new strain was found? The authors should highlight the new discovery of the current work in the section of abstract, discussion and conclusion. 

3. The gene background and fermentation mechanism of Candida magnolia should be added in the introduction part. What kind of genes that Candida magnolia may have that allow it transform the glucose to erythritol. And what reaction formula, what enzyme is involved in the fermentation process. These information should be added in the introduction part. 

Reviewer 2 Report

Overall comments: in general, the manuscript is well written, and the study is well designed. A few points need to be addressed before acceptance. See the specific comments below.

Specific comments:

1.       Line 78-79, the RSM has also been used to optimize the ethylene oxide microbial inactivation of Salmonella in cumin seeds:

Inactivation of Salmonella enterica and Enterococcus faecium NRRL B-2354 in cumin seeds using gaseous ethylene oxide.

2.       In Figure. 1, explain the different combinations of yeast extract and molasses having the erythritol yield. Which combination is the optimal one?

3.       Add x and y-axis labels for Fig. 2, 3, and 4.

Reviewer 3 Report

Overall, the manuscript entitled “Fermentative production of erythritol from cane molasses using Candida magnolia: Media optimization, partial purification, and characterization” was properly written and structured. However, I have some comments to author:

Line 17: Please, indicate which were the three most influential factors.

Lines 21-23: Why is a partial purification if it was shown to be 94% of purity? 

Lines 23-25: Is there another important information regarding the characterization?

Lines 85-86: Was the purity of the erythritol characterized? Please, revise.

Lines 3, 16, 63, 83, 90, etc.: The term “Candida magnoliae” must be homogenized. Some were written as “magnolia” or without cursive style.

Line 112 and 115: Please indicate the meaning of BSA and NAD, respectively.

Line 134: Please, could you provide each letter with each independent variable (i.e. A= molasses, B= yeast extract…) for better understanding. Indicate if did you refer to the concentration of molasses, yeast extract, KH2PO4 and MgSO4. Indicate the units from all the coded values from A to K. Also, improve the presentation of Table 1.

Lines 203-218: Add a reference from the method presented in section 2.9.

Lines 251-254: The same information was previously provided in lines 134-136. Please delete and correct.

Lines 263-265: Did “positively” and “negatively” mean that the erythritol formation increased and decreased, respectively? Please, clarify.

Lines 281-285: “was summarized” instead of “has been”. Use the past tense in lines 282-285, 302-305.

Lines 307-312: How do you explain that an increased concentration of KH2PO4 enhanced the erythritol yield?

Lines 318 and 319: Please, revise the sentence for better understanding.

Lines 327-328: So, what happen with the run 5 which is higher than 12.40%? Please, indicate.

Line 387: Please, could you revise the numbers, commas and units from the area under the curve. It seems that there is a mistake.

Line 405: Indicate what are figure 4a and 4b for better understanding.

Lines 423-424. In this section you must show the conclusion of your results, not the objective. Additionally, where did you assess the transition of kcal/g in molasses and erythritol? These results did not come from your research, you must delete.  

Lines 425-428: What those results imply? What is the contribution to the advance of the knowledge? Please, add this information.
